# The translation attenuating arginine-rich sequence in the extended signal peptide of the protein-tyrosine phosphatase PTPRJ/DEP1 is conserved in mammals

Luchezar Karagyozov[1]*, Petar N. Grozdanov[2], Frank-D. Böhmer[1]*

**1** Institute of Molecular Cell Biology, CMB, Jena University Hospital, Jena, Germany, **2** Department of Cell Biology & Biochemistry, School of Medicine, Texas Tech University Health Sciences Center, Lubbock, United States of America

* lucho3ka@yahoo.com (LK); i5frbo@uni-jena.de (FDB)

**Data Availability Statement:** All relevant data are within the manuscript and its Supporting Information files.

## Abstract

The signal peptides, present at the N-terminus of many proteins, guide the proteins into cell membranes. In some proteins, the signal peptide is with an extended N-terminal region. Previously, it was demonstrated that the N-terminally extended signal peptide of the human *PTPRJ* contains a cluster of arginine residues, which attenuates translation. The analysis of the mammalian orthologous sequences revealed that this sequence is highly conserved. The *PTPRJ* transcripts in placentals, marsupials, and monotremes encode a stretch of 10–14 arginine residues, positioned 11–12 codons downstream of the initiating AUG. The remarkable conservation of the repeated arginine residues in the *PTPRJ* signal peptides points to their key role. Further, the presence of an arginine cluster in the extended signal peptides of other proteins (E3 ubiquitin-protein ligase, NOTCH3) is noted and indicates a more general importance of this cis-acting mechanism of translational suppression.

## Introduction

After the start of translation, translation inhibition due to the interaction between the nascent polypeptide chain and the ribosome is reported [1, 2]. In eukaryotes, transient elongation arrest may be caused by consecutive prolines [3] or by an array of positively charged amino acids [4]. In a limited number of proteins, the signal peptides are longer than the canonical 20–25 residues; they consist of more than forty amino acid residues and contain an N-terminal extension and a hydrophobic sequence (h-region) far from the N-terminus [5]. This extension provides a convenient space for positioning of a translation attenuating amino acid sequence.

The human receptor-like protein tyrosine phosphatase, type J (PTPRJ, PTPReta, or DEP1) provides an example of the presence of a down-regulating sequence within the signal peptide. The human PTPRJ is a receptor-like protein tyrosine phosphatase of the R3 subtype characterized by an extracellular region, containing several tandem fibronectin type III (FNIII) domains, a single transmembrane region, and a single cytoplasmic catalytic domain [6]. In

**Funding:** The author(s) received no specific funding for this work.

**Competing interests:** The authors have declared that no competing interests exist.

human embryonic lung fibroblasts, the *PTPRJ* expression and activity were dramatically increased when cells approached confluence in comparison to sparse cells, suggesting a possible role in cell-density-dependent inhibition of proliferation. Thus, the name high cell density-enhanced phosphatase-1 or DEP1 was proposed [7]. *PTPRJ* is expressed in a variety of normal tissues, notably in hematopoietic cells (CD148 antigen), in epithelial tissues, including those of the digestive tract, and in the vascular endothelium [8]. Data suggest that *PTPRJ* is a tumor suppressor in different tissues. In mice, the gene encoding PTPRJ was mapped to a colon cancer susceptibility locus (Scc1) [9]. Negative regulation of the signaling of several receptor-tyrosine kinases (RTKs), including the epidermal growth factor receptor (EGFR) [10], the platelet-derived growth factor receptor [11], and Fms-like tyrosine kinase 3 [12] may be important in this context. A metabolic function of PTPRJ is indicated by the negative regulation of insulin receptor and leptin receptor signaling [13–15]. PTPRJ was also identified as an effective activator of Src-kinase in different cell types. This function of PTPRJ is important for platelet activation and thrombosis [16, 17], for efficient angiogenesis [18], and for regulating airway hyper-responsiveness [19].

Previous experiments [20] showed that: (1) the human *PTPRJ* transcripts predominantly initiate translation at the first AUG in a favorable context, numbered $AUG_{190}$. This results in a PTPRJ pre-protein with an N-terminally extended signal peptide and a hydrophobic signal sequence, which is far from the N-terminus. (2) The N-terminal extension contains an unusual arginine-rich cluster (RRTGWRRRRRRRR); its translation inhibits the overall PTPRJ expression.

To elucidate the importance of these findings it was of interest to examine the *PTPRJ* transcripts in mammals for sequences encoding the attenuating arginine-rich cluster. In the present paper, *PTPRJ* orthologs from placental mammals, marsupials, and monotremes were compared. The results revealed a similarity in the architecture of the *PTPRJ* transcripts. Several conserved features were noted: (1) uORFs are not present in the transcripts; (2) the *PTPRJ* transcripts encode signal peptides with N-terminal extension and a hydrophobic signal sequence which is rather distant from the N-terminus; (3) the extended signal peptides contain the attenuating arginine cluster. The remarkable evolutionary conservation of the attenuating sequence emphasizes the importance of suppressing the PTPRJ translation by the nascent peptide chain.

## Materials and methods

### Orthologous genes and data evaluation

The NCBI gene database (https://www.ncbi.nlm.nih.gov/gene) entry for the human protein tyrosine phosphatase receptor type J (*PTPRJ*, GeneID: 5795) was used to search for *PTPRJ* orthologs. The NCBI default routine designated 'NCBI's Eukaryotic Genome Annotation pipeline' was used, which employs the NCBI Gene dataset and a combination of protein sequence similarity and local synteny information (https://www.ncbi.nlm.nih.gov/kis/info/how-are-orthologs-calculated/). The search for orthologs was limited to mammals.

The 5′ end regions of the orthologous *PTPRJ* transcripts were examined for the presence of initiator AUG codons to determine the N-terminal end of the encoded proteins. The translation start in the mammalian *PTPRJ* transcript sequences was also predicted by the NetStart1.0 server at https://services.healthtech.dtu.dk/service.php?NetStart-1.0 [21]. The presence of a signal hydrophobic amino acid sequence (h-region) and the signal-peptidase cleavage sites were determined *in silico* using the SignalP 5.0 web server at https://services.healthtech.dtu.dk/service.php?SignalP-5.0 [22]. In the SignalP algorithm, the input sequence has an upper limit of 70 residues, thus, initially, the N-terminal amino acids were examined, and then—the

adjoining downstream region. The distribution of the elongating and initiating ribosomes in transcripts encoded by exon 1 (450 bp) of the human *PTPRJ* on chromosome 11 was visualized by the genome browser https://gwips.ucc.ie/ [23]. BLAST, PHI-BLAST (NCBI) and Clone Manager Suite 8 (Scientific and Educational Software) were used to search the database, compare and analyze the nucleotide and amino acid sequences.

## Protein-tyrosine phosphatase PTPRJ/DEP1 sequences

The analyzed 5′ end regions of the *PTPRJ* transcripts were from ten species: five placental mammals, four marsupials, and one monotreme.

**Placentals.**    Primates—*Homo sapiens* (human), mRNA transcript variant 1: NM_002843.4, protein: NP_002834.3

Rodents—*Mus musculus* (house mouse), mRNA transcript variant 1: NM_008982.6, protein: NP_033008.4

Cetaceans—*Delphinapterus leucas* (beluga whale), mRNA: XM_030764039.1, protein: XP_030619899.1

Ruminants–*Bos taurus* (cattle), mRNA: XM_024975918.1, protein: XP_024831686.1

Carnivore—*Enhydra lutris* (sea otter), mRNA: XM_022506988.1, protein: XP_022362696.1

**Marsupials.**    *Monodelphis domestica* (gray short-tailed opossum)—mRNA: XM_016422845.1, protein: XP_016278331.1

*Phascolarctos cinereus* (koala)—mRNA transcript variant X1: XM_021009664.1, protein: XP_020865323.1

*Sarcophilus harrisii* (Tasmanian devil)—mRNA transcript variant X1: XM_031941343.1, protein: XP_031797203.1

*Vombatus ursinus* (common wombat)—mRNA transcript variant X1: XM_027837689.1, protein: XP_027693490.1

**Monotremes.**    *Ornithorhynchus anatinus* (platypus)–mRNA transcript variant X2: XM_029060042.1, protein: XP_028915875.1

## E3 ubiquitin-protein ligase (ZNRF3) sequences

*Homo sapiens* (human), mRNA transcript variant 1: NM_001206998.2, protein: NP_001193927.1

*Mus musculus* (house mouse), mRNA transcript variant 1: NM_001080924.2, protein: NP_001074393.1

## Notch receptor 3 (NOTCH3) sequences

*Homo sapiens* (human), mRNA transcript: NM_000435.3, protein: NP_000426.2

*Mus musculus* (house mouse), mRNA: NM_008716.3, protein: NP_032742.1

# Results and discussion

## The *PTPRJ* orthologs in mammals

The NCBI gene database lists 123 mammalian orthologs of the human *PTPRJ*: placental mammals—118 orthologs, marsupials—4, monotremes—1. We analyzed five orthologs from major groups of placentals and all orthologs from marsupials and monotremes.

In some mammalian species, different splice variants encoding different PTPRJ isoforms are listed. We examined all isoforms for the presence of a hydrophobic region near the N-terminus. Only isoforms with a predicted cleavable signal peptide were analyzed further.

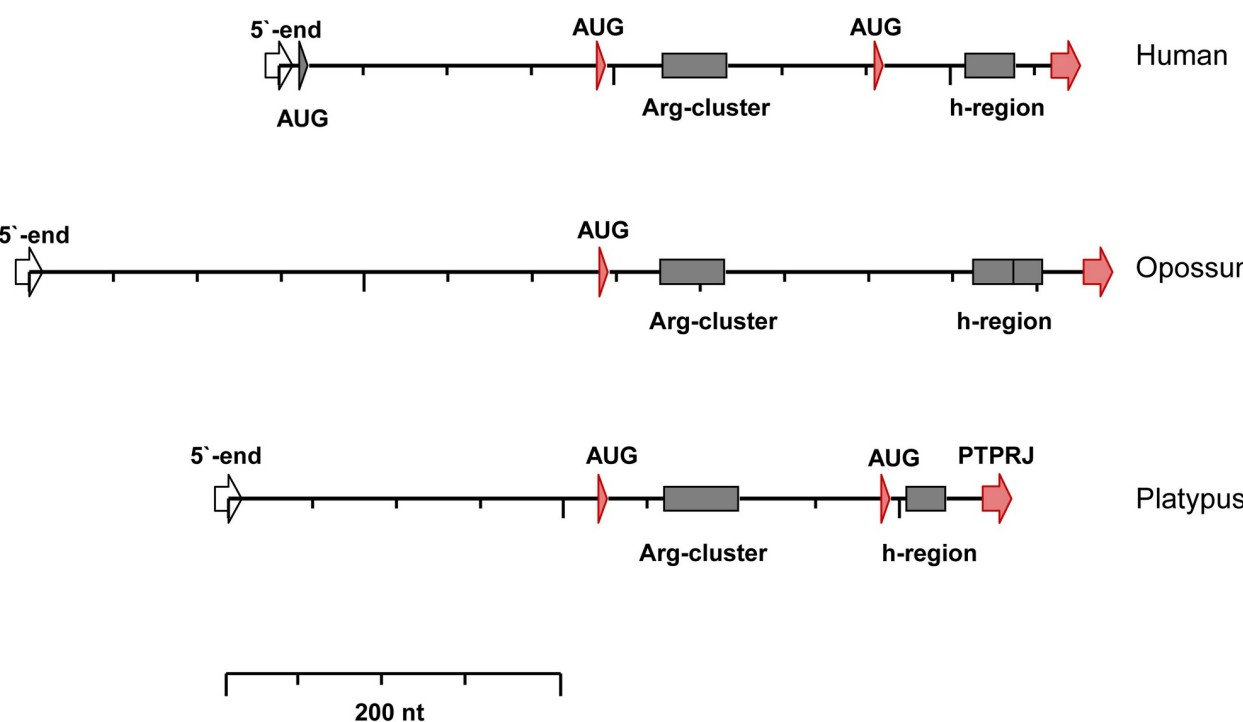

**Fig 1. Sequence pattern of the 5'-end of the mammalian *PTPRJ* transcripts.** The AUG codons are indicated. The position of the transcripts is adjusted to align the first initiating AUGs. Arg-cluster–repeated Arg residues; h-region–region of nonpolar amino acids; PTPRJ–the mature protein.

### The AUG initiator codons in mammalian *PTPRJ* transcripts

The arrangement of the AUG triplets at the 5′ end of the mammalian *PTPRJ* mRNA is shown in Fig 1. In all transcripts, the AUG codons are in-frame with the mature protein, with no intervening stop codons between them (see also S1, S3 and S5 Figs). Thus, no uORFs exist in the mammalian *PTPRJ* transcripts.

The number of the initiating AUGs differs between mammalian subdivisions; the placentals have three AUGs, the monotremes–two, and the marsupials–only one. Remarkably, the context of the single marsupial AUG is identical to the preferred starting site for translation in humans ($AUG_{190}$), which was identified previously [20].

It is firmly established that the nucleotides surrounding the AUG codons strongly affect initiation efficiency. Mutagenesis experiments with transfected COS cells [24] established an optimal context sequence for initiation (RCC**AUG**G, R is A or G). The nucleotides at positions -3 and +4 (the A in the AUG codon is +1) are of particular importance. The AUG context is categorized as strong (both crucial positions match the optimal sequence), favorable (only one match), or weak (no match at positions -3 and +4).

In mammalian *PTPRJ* transcripts, the nucleotide context of the AUG codons varies according to their scanning order (Table 1). In placentals, the AUG close to the 5' cap (6–15 nucleotides) is scanned first, however, it is in a weak context (CGC**AUG**A). The next AUG is in a favorable context with G at -3 and U at +4 (GCC**AUG**U); the context of the third AUG is also favorable, but with A at +4 (GCC**AUG**A). In monotremes, the context of both AUGs is favorable (GCC**AUG**U and GCC**AUG**A, respectively). The marsupials possess a single AUG, which is in a favorable context (GCC**AUG**U).

The AUG codons are arranged in 5′ to 3′ direction according to the movement of the scanning complexes.

**Table 1. Context of the AUG codons in the 5′ end of the *PTPRJ* transcripts.**

| Placentals. | | | |
|---|---|---|---|
| Organism | 1ˢᵗ AUG | 2ⁿᵈ AUG | 3ʳᵈ AUG |
| Consensus | CGC**AUG**A | GCC**AUG**U | GGC**AUG**A |
| **Monotremes.** | | | |
| Organism | 1ˢᵗ AUG | 2ⁿᵈ AUG | |
| Platypus | GCC**AUG**U | GGC**AUG**A | |
| **Marsupials.** | | | |
| Organism | 1ˢᵗ AUG | | |
| Consensus | GCC**AUG**U | | |

The optimal initiator sequence is RCC**AUG**G, R is A or G [24]. Critical positions are R at -3 and G at +4 (the A in the AUG codon is +1). The nucleotides surrounding the AUGs are from S1, S3 and S5 Figs.

To summarize, in all mammals, the AUG in a favorable context, at which the scanning 40S subunits arrive first, is with U in position +4. In placentals, the transcripts contain a preceding AUG in a weak context. In monotremes and placentals, the transcripts possess a downstream AUG in a favorable context, but with A in position +4. The functional significance of these differences is unknown. One may speculate that they reflect subtle differences in expression regulation.

To estimate additionally the potential of the different AUG codons to initiate translation the *PTPRJ* nucleotide sequences from placental mammals and platypus were submitted to the NetStart-1.0 web server. To predict translation start, this server takes into account a combination of local start codon context and global sequence information. Invariably the *in silico* evaluation gave the highest score to the AUG preceding the Arg-rich cluster.

## The efficiency of the AUG initiator codons in the human *PTPRJ* transcripts

In humans, there are three in-frame AUG codons positioned upstream of the hydrophobic region (Fig 1). In previous experiments, the efficiency of each of the human initiator codons was tested in reporter constructs [20]. Briefly, the outcomes were (1) when all three AUGs were mutated no reporter activity was detected; (2) the efficiency of the first initiator ($AUG_{13}$) was the weakest, and (3) the efficiency of the next two initiators ($AUG_{190}$ and $AUG_{355}$) appeared similar. Translation of the *PTPRJ* mRNA started predominantly at the first initiator in a favorable context ($AUG_{190}$).

These results are in agreement with published RiboSeq data (Fig 2). The sequences encoded by the first exon of the human *PTPRJ* do not support appreciable non-canonical translation initiation. The first initiator ($AUG_{13}$), which is in a weak context, is not "tight"; it leaks scanning complexes downstream towards the second initiator. According to the profiles of the initiating ribosomes, the third initiator ($AUG_{355}$) is not active.

## The mammalian *PTPRJ* transcripts code for signal peptides with N-terminal extension

The N-termini of the proteins targeted for secretion or membrane integration, usually harbor a short amino acid sequence–the signal peptide–instrumental for the translocation of the proteins into the ER membranes. In most cases, the signal peptide is 20–25 amino acids long. It contains a positively charged N-terminal region (1–5 residues), a hydrophobic region (h-region) (7–15 residues), and a signal-peptidase recognition site (1–5 residues) [25].

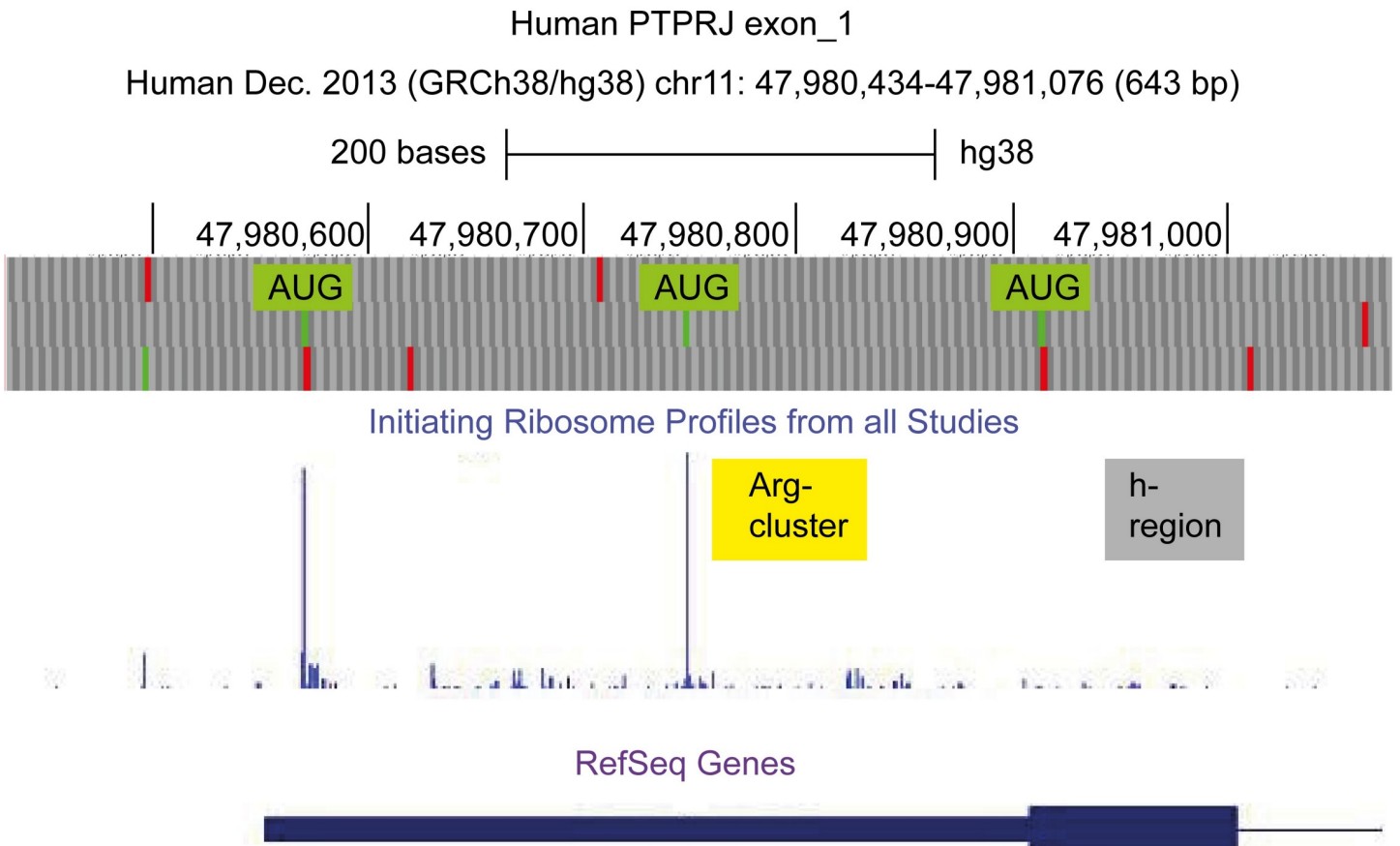

**Fig 2. Ribosome profiling of *hPTPRJ*, exon 1.** Top–the three reading frames with initiating codons (green) and stop codons (red). The AUGs in reading frame 2 are marked. Middle–RiboSeq data, initiating ribosomes. The positions of the Arg-cluster (yellow) and the region of nonpolar amino acids (h-region, grey) are indicated. Bottom—the RefSeq gene, exon 1.

In several proteins, however, the signal peptide is N-terminally extended and the hydrophobic region is far from the N-terminus [5]. Recent experiments showed that in *Plasmodium falciparum* the N-terminal extension of the signal peptides is irrelevant for their function [26].

The signal peptides of the mammalian *PTPRJ* proteins are with an N-terminal extension (Fig 1). In marsupials, the signal peptides are 94–103 residues long (S4 Fig). In the platypus (monotremes), the *PTPRJ* mRNA encodes two possible signal peptides. The translation launched from the first AUG results in long signal peptide of 76 amino acids (S6 Fig). In placentals, the AUG initiating codons are three. In humans, according to the profiles of the initiating ribosomes, the third initiator ($AUG_{355}$) is not active (Fig 2). Correspondingly, the first and–the presumably preferred—second initiators in placental mammals produce N-terminally extended signal peptides composed of 147–150 and 87–92 amino acid residues, respectively (S1 and S2 Figs).

## The arginine-rich sequence in the N-terminally extended signal peptides is highly conserved

The human *PTPRJ* transcripts encode an extended signal peptide, which contains repeated arginine residues (RRTGWRRRRRRRR). Earlier it was demonstrated that the arginine cluster

attenuates translation [20]. A comparison between mammalian orthologs revealed the presence of a similar sequence in the extended signal peptides of all mammals (Fig 3). Differences are minor. In placentals, the sequence of arginine residues is with three intervening amino acids (TGW/G). In marsupials, the string of arginine residues is interrupted by two amino acids (S/TW). In platypus, the arginine-rich cluster is 15 amino acids with one interruption (G). The arginine repeats in mammals are positioned 11–12 residues downstream of the initiating methionine.

The conservation of the composition and location of the arginine-rich cluster emphasizes the functional importance of these features. In the absence of uORFs, this seems to be a necessary mechanism to down-regulate PTPRJ expression.

## The mechanism of translation attenuation and potential biological significance

Previous experiments showed that: (1) the translation attenuation is not due to the presence of rare codons; (2) elimination of the repeated arginine residues by frame-shift mutations (plus and minus) is sufficient to up modulate *PTPRJ* expression [20].

Most likely, the inhibition of expression is due to the positive charge of the arginine residues. Stalling of ribosomes at positively charged residues, due to electrostatic interactions with the negatively charged exit tunnel was described in model experiments [4]. More recently RiboSeq data were interpreted to show that ribosomes in yeast and mammals stall at positively charged amino acids [27, 28]. The ribosome exit tunnel accommodates 30–40 amino acid residues [1, 2]. Therefore, it is reasonable to assume that in mammalian *PTPRJ* the translating ribosomes stall when the arginine residues of the nascent chain are in the exit tunnel. The result is translation inhibition. The high degree of conservation of this inhibitory sequence is a strong indication of its functional significance.

PTPRJ has numerous cellular substrates, such as RTKs, Src-family kinases, and others. It has been implicated in the regulation of a wide range of cellular functions. Enzymatic studies of the PTPRJ phosphatase domain revealed promiscuity with respect to substrate specificity and a very high intrinsic activity [29]. It appears therefore plausible that high levels of PTPRJ protein may disturb cellular functions or may even be toxic. Attenuation of PTPRJ translation by virtue of the here described features of the signal peptide may serve to prevent toxic effects and to allow fine-tuning of expression at the transcriptional level.

Other transcripts, encoding N-terminally extended signal peptides, may use a similar cis-acting mechanism to attenuate expression. An analysis of the extended signal peptides in the precursors of the human E3 ubiquitin-protein ligase ZNRF3 and of NOTCH3 revealed the presence of arginine clusters and stretches of proline residues. Both structures may cause ribosome pausing as the nascent chain is synthesized (Fig 4). In these cases, the efficiency of the nascent chains to throttle translation remains to be elucidated.

```
Placentals    1 MSPGKPGAGGAGTRRTGWRRRRRRRRRLEAATRAPGLGRTA
Marsupials    1 MSPGKPGAGGAERRRRSWRRRRRRRRPRPRPPAPA------
Monotreme     1 MSPGKPGAGETPPRRRRRRGRRRRRRRRRPQPGPATTKAAA
```

**Fig 3. Comparison of the arginine cluster in the N-terminally extended signal peptides in mammals.** Placentals and marsupials—consensus sequences, monotremes–platypus, see S2, S4 and S6 Figs. The initiating Met residues (green), the conserved Arg-residues (yellow) and the intervening amino acids (grey) are marked.

```
        E3 ubiquitin-protein ligase ZNRF3 isoform 1 precursor
Human   MRPRSGGRPGATGRRRRRLRRRPRGLRCSRLPPPPPLPLLLGLLLAAAGPGAARA▼KE
Mouse   MRPRSGGRPGAPGRRRRRLRRGPRG---RRLPPPPPLPLLLGLLLAAAGPGAARA▼KE

        Notch receptor 3 (NOTCH3)
Human   MGPGARGRRRRRRPMSPPPPPPPVRALPLLLLLAGPGAA▼AP
Mouse   MGLGARGRRRRRRLMALPPPPPPMRALPLLLLLAGLGAA▼AP
```

**Fig 4. The N-terminally extended signal peptide of the E3 ubiquitin-protein ligase ZNRF3 and NOTCH3 in humans and mouse.** The initiating Met residues (green), the Arg-cluster (yellow), the consecutive Pro residues (underlined) and the nonpolar signal sequences (grey) are shown. The signal peptidase cleavage site (•) was predicted *in silico* by the SignalP 5.0 Server.

## Supporting information

**S1 Fig. Alignment of the 5′ end of the *PTPRJ* transcripts encoding the extended signal peptides in placental mammals.**
(PDF)

**S2 Fig. Alignment of the extended signal peptides of PTPRJ in placental mammals.**
(PDF)

**S3 Fig. Alignment of the 5′ end of the *PTPRJ* transcripts encoding the extended signal peptides in marsupials.**
(PDF)

**S4 Fig. Alignment of the extended signal peptides of PTPRJ in marsupials.**
(PDF)

**S5 Fig. The 5′ end region of the *PTPRJ* mRNA in platypus encoding the extended signal peptide.**
(PDF)

**S6 Fig. The extended signal peptide of PTPRJ in platypus.**
(PDF)

## Acknowledgments

One of the authors (LK) highly appreciates the help of I. Stancheva for critically reading the manuscript and helpful suggestions.

## Author Contributions

**Conceptualization:** Luchezar Karagyozov, Frank-D. Böhmer.

**Investigation:** Luchezar Karagyozov, Petar N. Grozdanov.

**Methodology:** Luchezar Karagyozov, Petar N. Grozdanov.

**Supervision:** Frank-D. Böhmer.

**Writing – original draft:** Luchezar Karagyozov.

**Writing – review & editing:** Frank-D. Böhmer.

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
