## [Decision Letter · Decision Letter 0]

26 Oct 2020

PONE-D-20-30969

The translation attenuating arginine-rich sequence in the extended signal peptide of the protein-tyrosine phosphatase PTPRJ/DEP1 is conserved in mammals

PLOS ONE

Dear Dr. Frank D. Böhmer,

Thank you for submitting your manuscript to PLOS ONE. First of all, my personal apologies for the delay in getting back to you. Delay has been caused by the cancellation of 7 invitations to review the manuscript. After reading the manuscript and the single review I have decided to proceed with the process.

After careful consideration, we feel that it has merit but does not fully meet PLOS ONE’s publication criteria as it currently stands. Therefore, we invite you to submit a revised version of the manuscript that addresses all the minor the points raised during the review process.

We look forward to receiving your revised manuscript.

Kind regards,

Maria Gasset, Ph.D.

Academic Editor

PLOS ONE

Journal Requirements:

Reviewers' comments:

Reviewer's Responses to Questions

**Comments to the Author**

1. Is the manuscript technically sound, and do the data support the conclusions?

Reviewer #1: Yes

2. Has the statistical analysis been performed appropriately and rigorously? 

Reviewer #1: N/A

3. Have the authors made all data underlying the findings in their manuscript fully available?

Reviewer #1: Yes

4. Is the manuscript presented in an intelligible fashion and written in standard English?

Reviewer #1: Yes

5. Review Comments to the Author

Reviewer #1: The authors have previously (Karagyozov et al. 2008) presented experimental evidence that the human PTPRJ protein is N-terminally longer than previously thought, with translation starting predominantly from the second AUG in the mRNA, not the third. This results in an unusually long signal peptide of 90 amino acids.

In this manuscript, they strengthen this evidence by a bioinformatic analysis, showing that a functionally important arginine-rich stretch of sequence is widely conserved in mammals (both placentals, marsupials, and a monotreme).

When I look up human PTPRJ in UniProt (Q12913), I only see the "short" sequence, translated from the third AUG, with a predicted signal peptide of 35 amino acids. The authors should report their findings to UniProt.

Major comments:

1 - p.1 (and several other places in the manuscript): "... an N-terminal extension and a recessed hydrophobic signal sequence (Hiss et al., 2008)."; two comments:

* It is not clear what the authors mean by "recessed" - does it just mean that the hydrophobic region is far from the N-terminus? Hiss et al. do not use this term. Please define.

* It is not customary to refer to the hydrophobic part of a signal peptide as "signal sequence"; instead, it is normally called "h-region". This is also how Hiss et al. refer to it, while "Signal sequence" in Hiss et al. is used as a synonym for signal peptide. Please use "h-region" instead throughout the manuscript.

2 - p.2: "The NCBI gene database was used to search for PTPRJ orthologs in mammals": More detail is needed here. There are various ways of operationally defining orthology, so the authors should precisely describe how this was done.

3 - p.4: "The N-termini of the proteins targeted for secretion or membrane integration, harbor a short amino acid sequence – the signal peptide...": regarding proteins targeted for membrane integration, this is only true for some of them.

4 - p.5: "Recent experiments showed that the N-terminal extension of the signal peptides is irrelevant for the signal sequence function (Meyer et al., 2018)": it should be mentioned that Meyer et al's results did not concern mammals.

5 - Just a suggestion: I submitted the human nucleotide sequence (NM_002843.4) to the NetStart web server for prediction of start codons (https://services.healthtech.dtu.dk/service.php?NetStart-1.0), and indeed, AUG-190 scores higher than AUG-355 and AUG-13. Maybe the authors would like to do this for all their nucleotide sequences and see if this is a general observation?

Minor comments:

- on p.1, there is a reference to "Andersen et al., 2001", but Andersen et al. in the reference list is from 2005.

- p.2: "SignalIP" should be "SignalP"

- p.2: "one monotremes" should be "one monotreme"

6. PLOS authors have the option to publish the peer review history of their article (what does this mean?). If published, this will include your full peer review and any attached files.

Reviewer #1: No

---

## [Author Response · Author response to Decision Letter 0]

11 Nov 2020

Our point-by-point response to the comments of Reviewer #1 is below:

When I look up human PTPRJ in UniProt (Q12913), I only see the "short" sequence, translated from the third AUG, with a predicted signal peptide of 35 amino acids. The authors should report their findings to UniProt.

Thank you very much for this suggestion! We have submitted a request to UniProt for amendment of the Q12913 entry quoting our earlier work and the current manuscript. We hope very much that an appropriate alteration of the entry will be conducted in due time.

Major comments:

1 - p.1 (and several other places in the manuscript): "... an N-terminal extension and a recessed hydrophobic signal sequence (Hiss et al., 2008)."; two comments:

* It is not clear what the authors mean by "recessed" - does it just mean that the hydrophobic region is far from the N-terminus? Hiss et al. do not use this term. Please define.

We have now avoided throughout the manuscript using the descriptive term “recessed” and instead simply indicate that the hydrophobic region is distant/far away from the N-terminus.

* It is not customary to refer to the hydrophobic part of a signal peptide as "signal sequence"; instead, it is normally called "h-region". This is also how Hiss et al. refer to it, while "Signal sequence" in Hiss et al. is used as a synonym for signal peptide. Please use "h-region" instead throughout the manuscript.

We have altered the manuscript accordingly and use now throughout in text and figures the term “hydrophobic region” or “h-region” instead of “signal sequence”. 

2 - p.2: "The NCBI gene database was used to search for PTPRJ orthologs in mammals": More detail is needed here. There are various ways of operationally defining orthology, so the authors should precisely describe how this was done.

We describe now in more detail the procedure of finding human PTPRJ orthologs, that is genes in other organisms, which are similar to human PTPRJ due to common descent. The related text (p.4) spells now: ‘The NCBI gene database (https://www.ncbi.nlm.nih.gov/gene) entry for the human protein tyrosine phosphatase receptor type J (PTPRJ, GeneID: 5795) was used to search for PTPRJ orthologs. The NCBI default routine designated ‘NCBI's Eukaryotic Genome Annotation pipeline’ was used, which employs the NCBI Gene dataset and a combination of protein sequence similarity and local synteny information (https://www.ncbi.nlm.nih.gov/kis/info/how-are-orthologs-calculated/). The search for orthologs was limited to mammals.’

3 - p.4: "The N-termini of the proteins targeted for secretion or membrane integration, harbor a short amino acid sequence – the signal peptide...": regarding proteins targeted for membrane integration, this is only true for some of them.

We modified the sentence (now at p.11) to indicate that not all proteins targeted for membrane integration contain a signal peptide.

4 - p.5: "Recent experiments showed that the N-terminal extension of the signal peptides is irrelevant for the signal sequence function (Meyer et al., 2018)": it should be mentioned that Meyer et al's results did not concern mammals.

We indicated that this observation was made in Plasmodium falciparum (p.11).

5 - Just a suggestion: I submitted the human nucleotide sequence (NM_002843.4) to the NetStart web server for prediction of start codons (https://services.healthtech.dtu.dk/service.php?NetStart-1.0), and indeed, AUG-190 scores higher than AUG-355 and AUG-13. Maybe the authors would like to do this for all their nucleotide sequences and see if this is a general observation?

We followed this very valuable suggestion. The different AUG codons to initiate translation PTPRJ nucleotide sequences from placental mammals and platypus were submitted to the NetStart-1.0 web server. Indeed, the AUG positioned prior to the Arg-cluster has in all cases the highest score and presumably is most active in initiating translation. This is included in the text (p.9-10).

Minor comments:

We corrected all mistakes accordingly.

---

## [Decision Letter · Decision Letter 1]

24 Nov 2020

The translation attenuating arginine-rich sequence in the extended signal peptide of the protein-tyrosine phosphatase PTPRJ/DEP1 is conserved in mammals

PONE-D-20-30969R1

Dear Dr. Frank D. Böhmer,

We’re pleased to inform you that your manuscript has been judged scientifically suitable for publication and will be formally accepted for publication once it meets all outstanding technical requirements.

Kind regards,

Maria Gasset, Ph.D.

Academic Editor

PLOS ONE

Additional Editor Comments (optional):

Reviewers' comments:

Reviewer's Responses to Questions

**Comments to the Author**

1. If the authors have adequately addressed your comments raised in a previous round of review and you feel that this manuscript is now acceptable for publication, you may indicate that here to bypass the “Comments to the Author” section, enter your conflict of interest statement in the “Confidential to Editor” section, and submit your "Accept" recommendation.

Reviewer #1: All comments have been addressed

2. Is the manuscript technically sound, and do the data support the conclusions?

Reviewer #1: Yes

3. Has the statistical analysis been performed appropriately and rigorously? 

Reviewer #1: N/A

4. Have the authors made all data underlying the findings in their manuscript fully available?

Reviewer #1: Yes

5. Is the manuscript presented in an intelligible fashion and written in standard English?

Reviewer #1: Yes

6. Review Comments to the Author

Reviewer #1: (No Response)

7. PLOS authors have the option to publish the peer review history of their article (what does this mean?). If published, this will include your full peer review and any attached files.

Reviewer #1: **Yes: **Henrik Nielsen

---

## [Editor Report · Acceptance letter]

30 Nov 2020

PONE-D-20-30969R1 

The translation attenuating arginine-rich sequence in the extended signal peptide of the protein-tyrosine phosphatase PTPRJ/DEP1 is conserved in mammals 

Dear Dr. Böhmer:

I'm pleased to inform you that your manuscript has been deemed suitable for publication in PLOS ONE. Congratulations! Your manuscript is now with our production department. 

Kind regards, 

on behalf of

Dr. Maria Gasset 

Academic Editor

PLOS ONE